# Fingerprinting of Volatile Organic Compounds in Old and Commercial Apple Cultivars by HS-SPME GC/GC-ToF-MS

**DOI:** 10.3390/ijms252413478

**Published:** 2024-12-16

**Authors:** Kamil Szymczak, Justyna Nawrocka, Radosław Bonikowski

**Affiliations:** 1Institute of Natural Products and Cosmetics, Faculty of Biotechnology and Food Sciences, Lodz University of Technology, Stefanowskiego 2/22, 90-537 Lodz, Poland; radoslaw.bonikowski@p.lodz.pl; 2Department of Plant Physiology and Biochemistry, Faculty of Biology and Environmental Protection, University of Lodz, Banacha 12/16, 90-237 Lodz, Poland; justyna.nawrocka@biol.uni.lodz.pl

**Keywords:** apples, volatile organic compounds, fingerprinting

## Abstract

Flavor is the most important feature consumers use to examine fruit ripeness, and it also has an important influence on taste sensation. Nowadays, more and more consumers pay much attention not only to the appearance but also to the fruit’s aroma. Exploiting the potential of headspace solid-phase microextraction (HS-SPME) combined with sensitive two-dimensional gas chromatography and the time-of-flight mass spectrometry (GC/GC-ToF-MS) method within 30 old/traditional cultivars of apples (*Malus domestica* Borkh) coming from the same germplasm and 7 modern/commercial cultivars, 119 volatile organic compounds (VOCs) were identified. The largest group was esters (53), followed by alcohols (20), aldehydes (17), ketones (10), and acids (10). The richest volatile profile was ‘Grochówka’, with 61 VOCs present. The results revealed a visible difference based on VOC levels and profiles between the different apple cultivars, as well as visible similarities within the same cultivar coming from different farms. Based on a PCA, the commercial cultivars were separated into 7 clusters, including (1) ‘Gala’, (2) ‘Melrose’, (3) ‘Red Prince’, (4) ‘Lobo’, (5) ‘Ligol’, and (6) ‘Szampion’. The results of this study indicate that the profile of volatile compounds may be a useful tool for distinguishing between commercial and old apple cultivars, as well as for the varietal classification of apples from different locations. The developed method can also be used to identify other fruit varieties and origins based on their VOC composition. This may prove to be particularly valuable in the case of establishing a Protected Designation of Origin or Protected Geographical Indication.

## 1. Introduction

Apples (*Malus domestica* Borkh) are one of the most often cultivated and consumed fruits worldwide. Considering the annual production, only watermelons and bananas are more popular [1]. Poland is the biggest apple exporter in the world and the fourth largest producer of apples, right after China, Turkey, and the USA. Over 7500 apple cultivars are already known [2], and some authors estimate that even tens of thousands can be found all over the world [3]. However, the vast majority of known cultivars do not have economic significance. Throughout the years of selective breeding and genetic modification, the aim was to obtain apple cultivars that would meet consumers’ as well as producers’ expectations. For the majority of consumers, the most important aspect is an attractive look. On the other hand, for producers, the ultimate apple cultivar should be immune to diseases, mature quickly, endure the conditions of transshipment and transport after harvesting, and have a long shelf life, meaning the time it can be on a store shelf with no signs of aging. The best example is the ‘Red delicious’ cultivar, which was considered one of the less tasty apples (described as uncomfortably dry). However, for many years, it was the most popular variety in the USA because of its perfect look and because it turned red before any other cultivar, so it could be picked earlier and stored longer. Yet no one considers how these breeding programs have a strong impact on volatile organic compound profiles (aroma), which are strongly associated with taste sensation [4,5]. Aroma has always been the basis for people to assess the suitability of food for consumption. A pleasant, sweet aroma suggests that the fruit is already ripe and will be tasty, sweet, and therefore nutritious. The newest research evidenced that approximately 1/3 of perceived sweetness can be explained by the appropriate composition of volatile organic compounds (VOCs) [6].

In the last decade, there has been growing interest in food products produced in an ecological and environmentally friendly way, including fruits such as apples. In particular, attention is paid to the ancient cultivars of apples that are more resistant to diseases or have improved sensory properties compared to modern, commercially grown cultivars [7]. Moreover, regional and classic cultivars are a particular crop gene pool that needs to be preserved and promoted as a gene bank to maintain biodiversity.

In apples, there is an extremely high number of up to 500 different VOCs [8]. The composition of apple VOCs has been the subject of numerous studies (Table 1). Some studies focused mainly on comparing as many varieties as possible (42 varieties in [2] and 40 in [9]), while other researchers tried to determine as many compounds as possible (498 in [8] and 399 in [10]). Taking into account the diversity of varieties and the number of compounds determined in them, this is still an insufficient amount of knowledge. As far as extraction and analysis methods are concerned, it can be generally assumed that the method using solid-phase microextraction (SPME) is the most sensitive and the most developed. Additionally, the quality of analyses is improved by the use of two-dimensional gas chromatography coupled with time-of-flight mass spectrometry (GC/GC-ToF-MS).

The literature provides the frequency of 23 aroma components, mainly esters, which have a significant impact on the sensory description of the aroma of apples [17]. As indicated in Table 2, the most important compounds that affect the aroma of apples differ in terms of odor detection threshold by several orders of magnitude. In addition, the smell of some of them is not at all similar to the smell of apples. Also, in [17], it was pointed out that there is no key characteristic compound for any given cultivar.

Numerous studies indicate significant differences in the VOC profiles between varieties [2,9,14,16]. Therefore, in this study, we did not use fingerprinting to differentiate varieties as was shown in the literature [19,20]. The aim of our analyses was to check if it is possible to fingerprint apple cultivars grouped according to the same varieties by their VOC profile. The genetic background is indicated as the most important factor responsible for the differences in chemical composition [4,21]. However, it is noteworthy that many other factors, such as production region, climatic, atmospheric, and soil conditions [22], orchard management practices [23], or even the harvesting method [24], also play an important role.

The subject of this study was 37 old and new apple varieties, both from one germplasm and from different orchards, which gave a total of 87 individual samples. Since fingerprinting from the available literature focused on demonstrating differences between varieties, we decided to check whether it was possible to demonstrate similarities within varieties originating from different areas.

## 2. Results and Discussion

The analysis of volatile compounds in plant-based products, including fruits and vegetables, is a valuable tool in both scientific research and the food industry, with applications spanning a range of purposes. In the context of fruit, including apples, the characterization of volatile compounds has been primarily employed as a means of identifying and differentiating between various food products derived from them, including ciders and other beverages [25], as well as being utilized as a parameter for assessing the processing, storage, and ripening of fruits [19,20]. Recently, efforts have been made to identify and characterize volatile compounds in apples with regard to their potential use in cultivar classification and fingerprinting [16].

In the present study, the results of peak identification showed the presence of 119 VOCs identified (Appendix A in Appendix A). The largest group was esters (53), followed by alcohols (21), aldehydes (16), ketones (10), and acids (10). The richest volatile profile was ‘Grochówka’, with 61 VOCs present (Table 3). The least complex profile was ‘Kosztela’, with 24 compounds.

According to Dixon and Hewett [17], there are 23 important apple volatile compounds affecting aroma. A total of 19 of them are present in our samples, and only 4 of them were not detected.

Old cultivars were characterized by a higher number of detected compounds (39.3 on average) compared to modern cultivars (35.1). These findings are in alignment with the results of the research conducted by Ciesa et al. [19], which demonstrated a greater VOC chemodiversity in old apple cultivars compared to modern cultivars, probably due to their greater genetic variability. Interestingly, in the present study, when calculated by the mean mass of volatile compounds, the commercial cultivars had significantly more of them (27.53 mg/kg on average) compared to the old cultivars (18.14 mg/kg). It is also worth noting that there is no statistically significant difference in the mean number of alcohols, ketones, or acids present; however, a significantly higher number of aldehydes is observed in the old varieties than in the modern ones. It can, therefore, be said that statistically, although the commercial cultivars have more scent (by mass), the old ones have richer profiles. A comparison of the main volatile organic compounds (VOCs) content in selected apple cultivars is presented in Table 4, and a full table (Appendix A) with all the detected compounds, analyzed cultivars, and statistics is available in the Appendix A.

Another interesting aspect concerned the comparison of our quantitative results with those obtained by other groups of researchers. In two relatively recent studies, researchers compared 35 and 40 apple cultivars, determining 39 and 78 VOCs, respectively [9,16]. Wu et al. [16] determined the esters in the maximum amount of 28,307.42 µg/L, alcohols up to 183,500.00 µg/L, aldehydes up to 162,032.74 µg/L, and ketones up to 2204.98 µg/L using 2-octanol as an internal standard. In turn, Yang et al. [9] determined the esters in the maximum amount of 10,087.55 µg/kg, alcohols up to 542.07 µg/kg, aldehydes up to 4435.22 µg/kg, and ketones up to 44.64 µg/kg using 3-nonanone as an internal standard. These values differ significantly from each other and also differ from the results in a recent study. We determined the esters in the maximum amount of 22,553.1 µg/kg, alcohols up to 4829.3 µg/kg, aldehydes up to 3747.6 µg/kg, and ketones up to 1306.9 µg/kg. Most likely, the discrepancies result from the different standards used for quantitative determinations. In another paper, researchers determined VOCs from only one apple variety based on cyclohexanone as a standard [26]. The results of quantitative analyses showed the presence of esters in the maximum amount of 48,705.41 µg/kg, alcohols up to 25,578.92 µg/kg, aldehydes up to 31,862.49 µg/kg, and ketones up to 1188.32 µg/kg. Although, in this case, the results may be more influenced by the fact that the variety itself probably has a different profile than others, the selection of the standard (ketone) itself was probably not without influence. We can assume that depending on the chemical group of the standard used, diametrically different response factors are observed in the chromatograms and, consequently, the quantitative values of the analysis. We are aware that quantitative analyses of each compound separately for hundreds of compounds present in the samples are impossible. However, we believe that in order to maintain the reliability of the results, at least one calibration curve/one internal standard should be performed for each of the main analyzed groups of compounds, i.e., esters, alcohols, aldehydes, ketones, and acids.

Regarding the evaluation of whether it is possible to fingerprint apple cultivars or at least find some key characteristics allowing for a distinction between cultivars, two-level statistical analyses were performed. Based on the volatile profile, the preliminary analysis, which separately included all 119 VOCs identified in apples, allowed the old and commercial cultivars to be separated (Figure 1 and Figure 2). The only exceptions were ‘Koksa Pomarańczowa’ and ‘Kosztela’ apples from one cultivar, which were matched to commercial cultivars, and ‘Lobo’ apples from two cultivars, which were matched to the old cultivars.

The second-level PCA was conducted using the raw data of apples belonging to commercial cultivars in order to identify the groups of compounds that best represent the variance of the results (Figure 3).

Based on the 12 components generated with 70 compounds present in the apples of the commercial cultivars, the cultivars were separated (Figure 3, Appendix A in Appendix A). The varieties were grouped into 7 clusters: (1) ‘Gala’, (2) ‘Melrose’, (3) ‘Red Prince’, (4) ‘Lobo’ without ‘Lobo 7’ samples, (5) ‘Ligol’, (6) ‘Szampion’, and (7) only ‘Lobo 7’ samples. A Kruskal–Wallis test and the median test allowed for choosing 34 representative compounds that significantly differentiated the mentioned cultivars (Figure 3, Appendix A in Appendix A). Based on the obtained results, it may be assumed that the volatile profile has a good potential to be used to fingerprint apple cultivars.

Considering the genetic diversity of apple varieties [4,7], which is a direct reason for the differentiation of VOC profiles, it is particularly interesting how well samples from different crops within a variety match each other, especially for such varieties as ‘Gala’, ‘Ligol’, or ‘Szampion’. It is also worth noting that the geographical factor (location of the orchard) is of lesser importance because otherwise, the samples would be grouped by number, not cultivar. In this case, it would be appropriate to analyze samples from further locations or even other countries to confirm how climate affects the composition of VOCs. On the other hand, taking into account the number of crossbreeds of varieties that are often combined under one name, it may explain why varieties such as ‘Gala’ or ‘Koksa pomarańczowa’ match much worse.

A statistical analysis was conducted to identify a set of compounds that could be considered characteristic of a given cultivar. The ‘Gala’ cultivar was found to have a notable abundance of 2,3-Dihydrofuranone, Estragole, and Hex-1-en-5-yl acetate and an absence of Propyl butyrate, Oct-1-en-3-ol, Butyl 2-methylpropanoate, and Hept-3-en-6-ol. The ‘Melrose’ cultivar was characterized by a high content of Propyl butyrate, Butyl 2-methylbutyrate, Hex-2-enal, 2-Methylbutyl acetate, and Hexyl hexanoate and an absence of Pent-1-en-3-one, 6-Methylhept-5-en-2-ol, Benzaldehyde, Hexyl octanoate, Butyl 2-methylpropanoate, Octanoic acid, Hept-3-en-6-ol, 2,3-Dihydrofuranone, 2-Ethylhexanol, and Hex-1-en-5-yl acetate. The ‘Red Prince’ cultivar was found to have a notable abundance of Hexyl 2-methylbutyrate, Butyl propanoate, and Hexyl acetate and an absence of Propyl butyrate, Benzaldehyde, Butyl 2-methylpropanoate, Octanoic acid, Hept-3-en-6-ol, 2-Ethylhexanol, and Hex-1-en-5-yl acetate. The following characteristics are unique to ‘Lobo’: a high content of Heptan-2-ol and an absence of Pent-1-en-3-one, Hexyl octanoate, Butyl 2-methylpropanoate, and Hex-1-en-5-yl acetate. The ‘Ligol’ cultivar includes a high content of Hept-2-enal, Heptan-2-ol, Farnezen (sum of isomers), Hexyl butyrate, and Hexyl octanoate and does not include Pent-1-en-3-one, Propyl butyrate, Butyl 2-methylbutyrate, 2-Methylbutyl acetate, Propyl acetate, and Hex-1-en-5-yl acetate. The ‘Szampion’ cultivar was found to have a notable abundance of Pent-1-en-3-one, Hept-2-enal, Pentanol, Butyl 2-methylpropanoate, and Hex-1-en-5-yl acetate and an absence of Butyl 2-methylbutyrate, Heptan-2-ol, 2-Methylbutyl acetate, Propyl acetate, and Hept-3-en-6-ol. The aforementioned groups of compounds were initially classified in accordance with the specified cultivars.

It can be assumed that similar fingerprint analyses can also be performed in other fruits with complex VOC profiles, such as pears [21,27], plums [28,29], or quinces [30], and they can probably be used to distinguish between varieties. More and more attention is being paid to the authentication of the origin of products. In the case of awarded Protected Designation of Origin or Protected Geographical Indication labels, it is particularly important to be able to reliably and confidently determine the characteristic features of agricultural products on the basis of which labels were awarded. The developed method works very well in the case of apples, and we can conclude that it will be equally effective in the case of other fruits with a complex VOC composition, as well as many other agricultural and food products.

## 3. Materials and Methods

### 3.1. Fruit Samples

The study was carried out on apples belonging to 30 different old cultivars (‘Boskoop’, ‘Dean’s Codlin’, ‘Galloway Pippin’, ‘Grafsztynek Inflancki’, ‘Grochówka’, ‘Jakub Lebel’, ‘James Grieve’, ‘Kalwila Aderslebeńska’, ‘Kantówka Gdańska’, ‘Koksa Pomarańczowa’, ‘Kosztela’, ‘Kronselska’, ‘Krótkonóżka Królewska’, ‘Książę Albert’, ‘Książę Albrecht Pruski’, ‘Malinowa Oberlandzka’, ‘Niezrównane Peasgooda’, ‘Pepina Linneusza’, ‘Pepina Ribstona’, ‘Piękna z Rept’, ‘Reneta Blenheimska’, ‘Reneta Harberta’, ‘Reneta Kanadyjska’, ‘Reneta Kulona’, ‘Reneta Strauwalda’, ‘Reneta z Brownlee’, ‘Schieblers Taubenapfel’, ‘Szara Reneta’, ‘Złota Reneta’, and ‘Złotka Kwidzyńska’) that were grown in the experimental fields of the Research Institute of Horticulture in Skierniewice, Poland. In addition, we examined the VOC profiles of 7 commercially grown cultivars: ‘Gala’, ‘Golden Delicious’, ‘Melrose’, ‘Lobo’, ‘Ligol’, ‘Red Prince’, and ‘Szampion’. Each of those cultivars comes from local farmers from 3 to 9 various orchards located in Łódź Voivodeship. Modern varieties marked with the same number mean that they come from the same farm. In total, the analyses covered 87 individual apple samples, each in three repetitions. Analyses were carried out on freshly harvested apples to avoid the possible effect of prolonged storage on the volatile profile of the fruits.

### 3.2. Chemicals

Methanol, sodium chloride, analytical standards, and standards for calibration curves were purchased from Sigma–Aldrich (Steinheim, Germany). The SPME fibers assembly in a 23Ga needle with a 50/30 µm divinylbenzene/carboxen/polydimethylsiloxane (DVB/CAR/PDMS) coating was obtained from Supelco (Bellefonte, PA, USA). We purchased 20 mL screw cap vials from Kinesis (Altrincham, UK).

### 3.3. Sample Preparation and Extraction

Sample preparation and extraction were performed according to the method described in [10] with some modifications. Apples were rinsed in distilled water and drained to dryness. Then apple cores were removed, fruit tissue cut into small pieces, and 100 g of each sample was ground with 30 g of sodium chloride. The addition of NaCl made grinding easier and prevented enzymatic reactions from occurring and, at a later stage when it dissolves, supports the release of volatile compounds from the aqueous phase. In the next step, 50 mL of distilled water was added to the sample and homogenized within 3 min. Then 10 g of homogenate was moved to the SPME glass vial and screwed up tightly. The vial was moved to an ultrasound-assisted water bath at 30 °C for 20 min. After that time, the sample was put under immediate SPME extraction.

Standard solutions containing pentanol, 2-methylpentanal, 4-methylpentanoic acid, 2-methylheptan-3-one, and hexyl acetate were prepared as a methanolic mix. Volume ranges of 1 µL to 600 µL (depending on the compound) were diluted in 10 mL of methanol. For calibration curves, 10 mL distilled water and 2.5 g of NaCl were added to 20 mL screw-cap SPME vials and spiked with methanolic standards solution.

### 3.4. SPME Extraction

SPME fibers with a DVB/CAR/PDMS coating were used. Due to SPME fiber coating lifetime, extraction efficiency, and repeatability of analyzed samples, SPME fibers were conditioned with manufacturer recommendations and used in headspace extraction mode for both the apple samples and standards’ solutions. Incubation and extraction were performed at 30 °C for 15 and 60 min, respectively, with a 500 rpm agitation speed. Then, desorption was performed for 10 min at 240 °C in the splitless mode.

### 3.5. GC/GC-MS Method

Headspace SPME and GCxGC-ToF-MS analyses were performed on the LECO Pegasus 4D apparatus, equipped with the Agilent 6890N GC, a high-speed ToF mass spectrometer (LECO, St. Joseph, MI, USA), and a MultiPurpose Sampler (MPS 2) autosampler (Gerstel GmbH, Mulheim an der Ruhr, Germany). The column set was 5% phenyl and 95% polysilphenylene-siloxane BPX5 (30 m × 0.25 mm × 0.25 µm) capillary column (SGE Analytical Science, Melbourne, Australia) in the first dimension, coupled to medium-polar 50% phenyl polysilphenylene-siloxane BPX50 (2 m × 0.1 mm × 0.1 µm) (SGE Analytical Science, Melbourne, Australia) in the second dimension. Helium was the carrier gas at a flow rate of 1.5 mL/min for the entire run. The injector performed in the splitless mode. Modulation parameters consisted of a 6 s modulation period (1.6 s hot pulse time and 1.2 s cool time between stages) and a modulator temperature offset of 15 °C relative to the secondary oven temperature. The GC oven temperature was programmed at 35 °C for 5 min, then increased at 3 °C/min up to 245 °C, and held for 3 min (total time 78 min), while the secondary oven programming offset was 5 °C above the primary oven. The mass spectrometer operated in the Electron Impact mode at −70 eV and in a scan range (*m*/*z*) from 33 to 550 amu, with an ion source at 220 °C and a transfer line at 250 °C.

### 3.6. Automated Data Processing

Chromatograms were analyzed by automated ChromaTOF-GC Software (version 4.44) data processing software. Automated library searching was based on the National Institute of Standards (NIST MS Search, version 2.0) and Wiley 8 mass spectral libraries. The signal-to-noise (S/N) threshold for peak finding was set at a relatively high level of 500 to avoid numerous peaks after the deconvolution process or with low mass spectral similarity. Retention indices (RIs) were calculated from the retention times of a series of *n*-alkanes with linear interpolation.

The correctness of the automated identification of compounds to peaks was verified based on mixtures of standards, retention indices, and our own compound databases [31].

### 3.7. Statistical Analyses

MS Excel, Statistica version 13.1, and GradeStat version 2.5 were used for statistical analyses. Depending on the statistical method, the analyses were performed on raw data or on data that had been normalized to range [0, 1] in accordance with the following formula:$$z=\frac{x-min(x)}{\left[\max\left(x\right)-\min\left(x\right)\right]}$$

In order to ascertain whether it is feasible to initially differentiate between apples belonging to old and commercial cultivars, the initial analysis was conducted on normalized data, encompassing separately all 119 VOCs identified in apples. To more effectively demonstrate the distinctions between the cultivars, a gradation analysis was conducted, and the findings were represented using a heatmap. The preliminary cluster analysis was conducted using the Ward method combined with Euclidean distance.

The second-level analysis was conducted using apples belonging to commercial varieties. A principal components analysis (PCA) was performed with the objective of identifying the key compounds that would enable the differentiation of apples belonging to disparate cultivars. A total of 70 compounds that were present in apples of the commercial varieties were subjected to analysis. Based on the Kaiser criterion, the number of components in the analysis was reduced to 12, which allows for a description of the data set at the level of 80.9%. Cluster analysis and non-parametric analysis of variance were performed on the new data sets. ANOVA was performed using the Kruskal–Wallis test and the median test. Based on the demonstration of significant differences at the *p* < 0.05 level, representative compounds were selected to generate a hit map for the commercial apple cultivars.

## 4. Conclusions

This study was carried out to characterize the volatile profiles of 30 different old apple cultivars and 7 commercial ones. The chemical composition of apples has been reported in several articles. Depending on the method, sensitivity, and aim of the study, the twelve investigations reported from 30 up to almost 500 different VOCs that various apple cultivars can contain (Table 1). The current study shows that although a relatively high threshold level at the data processing stage was used, a significant number of compounds were identified. Moreover, each cultivar shows a distinctive VOC pattern, and there were even visible similarities within the same cultivar coming from different farms. The results show the potential for a database for the fingerprinting of apples based on their aroma profiles.

To our knowledge, this is the first attempt at apple fingerprinting as a means of distinguishing cultivars based on the composition of the main volatile compounds in fresh fruits. The results of this study indicate that the profile of volatile compounds may be a useful tool for distinguishing between commercial and old apple cultivars, as well as for the varietal classification of apples from different locations. Many other fruits with complex VOC patterns, such as pears, plums, or quinces, can also be identified by varieties using the developed method.

## Figures and Tables

**Figure 1 ijms-25-13478-f001:**
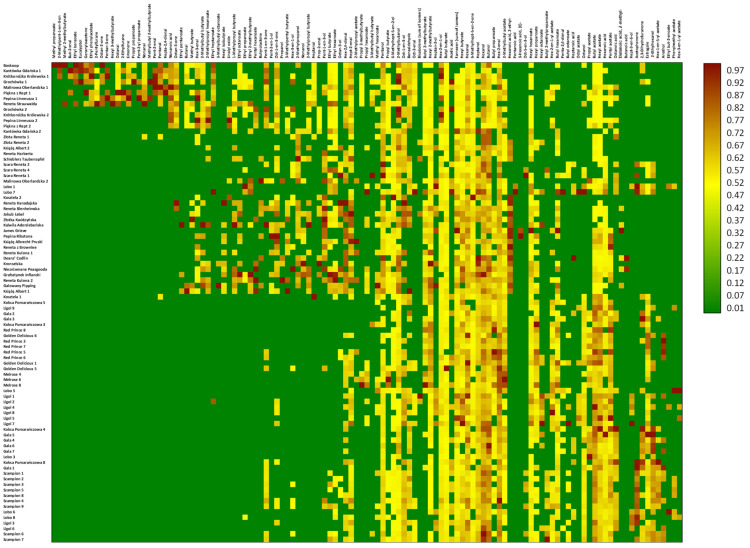
The preliminary heatmap analysis encompassing separately all 119 VOCs identified in apples, performed with old and commercial cultivars. The figure presents the results after gradation analysis.

**Figure 2 ijms-25-13478-f002:**
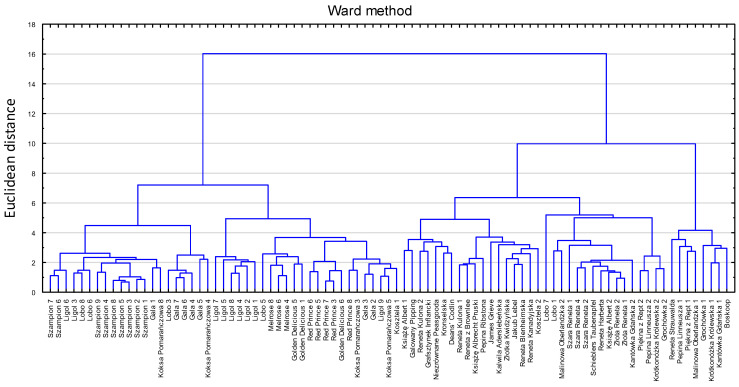
The preliminary cluster analysis encompassing separately all 119 VOCs identified in apples, performed with old and commercial cultivars. The Ward method combined with Euclidean distance was applied.

**Figure 3 ijms-25-13478-f003:**
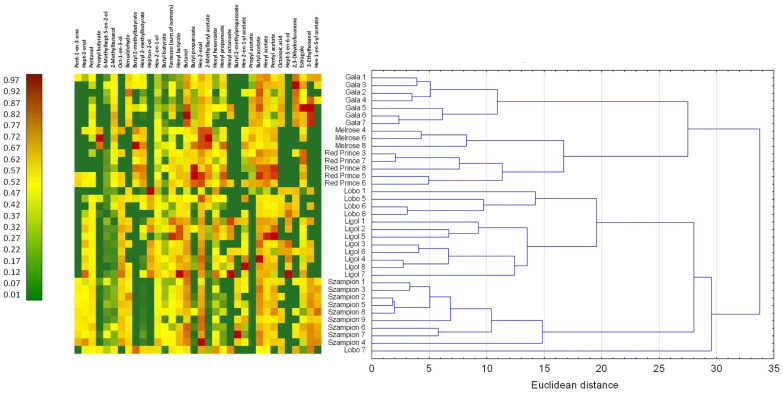
The second-level heatmap with selected compounds with the strongest influence on apple cultivar differentiations and the cluster analysis encompassing 12 components after a PCA was performed.

**Table 1 ijms-25-13478-t001:** Insight through selected publications on apple VOCs.

Number of Identified VOCs	Technique	Number of Cultivars	Year of Publication	Literature
498	HS GC-MS	1	2004	[8]
33	HS GC-MS	1	2006	[11]
30	HS	1	2007	[12]
100	SPME GC-MS	3	2009	[13]
72	SPME GC-MS	18	2012	[14]
399	SPME GC/GC-MS	1	2012	[10]
69	SPE GC-MS/MS	5	2014	[15]
118	HS-SPME GC-MS	42	2017	[2]
95	SPME GC-MS	17	2017	[6]
78	HS-SPME GC-MS	40	2021	[9]
39	HS-SPME GC-MS	35	2022	[16]
119	HS-SPME GC/GC-MS	37	Current study	

**Table 2 ijms-25-13478-t002:** The most important organic volatile compounds that have a significant impact on the aroma of apples [17,18].

	Compound	Flavor	Odor Detection Threshold [ppb]
Aldehydes	Acetaldehyde	pungent, fresh, aldehydic, refreshing, green	15–120
	2-Hexenal	sweet, almond, fruity, green, leafy, apple, plum, vegetable	17
	Hexanal	green, woody, vegetative, apple, grassy, citrus	4.5–5
Alcohols	Butanol	oily, sweet, balsamic	500
	Hexanol	green, herbaceous, woody, sweet	2500
	2-Hexenol	sharp, green, leafy, fruity, unripe banana	70
Esters	Butyl acetate	sharp, ethereal, fruity, banana	2
	Pentyl acetate	ethereal, fruity, banana, pear, apple	15
	Hexyl acetate	fruity, green, apple, banana, sweet	2
	2-Methylbutyl acetate	sweet, banana, fruity, ripe, estery, tropical	22
	Ethyl butyrate	strong, ethereal, fruity, banana, pineapple	1
	Ethyl 2-methylbutyrate	fruity, fresh, berry, grape, pineapple, mango, cherry	0.1–0.3
	Estragole	sweet, phenolic, anise, harsh, spice, green, herbal, minty	10
	Methyl 2-methylbutyrate	ethereal, fruity, green, sweet	0.25
	Propyl 2-methylbutyrate	winey, fruity, apple, pineapple	7
	Butyl 2-methylbutyrate	fruity, tropical, green, ethereal, herbal, celery, cocoa, peach, grassy	17
	Hexyl 2-methylbutyrate	green, waxy, fruity, apple, banana, woody	22
	Butyl hexanoate	fruity, pineapple, berry, apple, juicy, green, winey, waxy	250
	Hexyl propanoate	pear, green, fruity, musty, rotting	8
	Butyl butyrate	fruity, banana, pineapple, green, cherry, tropical, ripe fruit	100
	Butyl propanoate	fruity, sweet, banana, tropical, tutti-frutti	25–200
	Hexyl butanoate	green, sweet, fruity, apple, waxy	250
	Hexyl hexanoate	green, sweet, waxy, fruity, tropical, berry	unknown

**Table 3 ijms-25-13478-t003:** The presence of compounds in old and commercial cultivars.

	Old Cultivars			Commercial Cultivars			All
	Min	Mean	Max	Min	Mean	Max	
Mass of VOCs [mg/kg]	6.18	18.14	50.52	13.94	27.53	44.49	-
Number of VOCs detected	24	39.3	61	26	35.1	44	119
Number of esters	4	16.8	27	8	14.8	19	53
Number of alcohols	5	9.5	15	6	9.7	13	20
Number of aldehydes	2	6.1	14	2	4.3	7	17
Number of ketones	0	3.0	8	0	2.1	4	10
Number of acids	1	2.1	4	0	2.0	5	10
Number of other VOCs	0	1.8	6	1	2.3	4	9

**Table 4 ijms-25-13478-t004:** Comparison of main volatile organic compounds (VOCs) content in selected apple cultivars. Values in µg/kg of fresh fruit. A full table with all the detected compounds, analyzed cultivars, and statistics is available in the Appendix A.

Compound Name	Boskoop	Galloway Pippin	Golden Delicious 1	Grafsztynek Inflancki	Grochówka	Jakub Lebel	Kantówka Gdańska	Kosztela	Kronselska	Krótkonóżka Królewska	Książę Albert	Melrose 6	Red Prince 5	Szampion 3	Szara Reneta 1
2-Methylbutyl acetate		1379.3	1823.7	828.7	363.5			1453.3	552.1		298.1	7850.7	3878.1	253.0	
2-Methylbutanol	702.2	906.5	774.4	338.1	1539.3		231.0	252.4	1089.3	1384.2	968.0	532.8	999.0	274.6	627.1
2-Methylbutyl butyrate	75.6	616.3			128.0		109.5		81.8	52.6					
2-Methylpropanol	188.2	239.6		307.1		132.2	75.5		313.5						164.8
6-Methylhept-5-en-2-ol		317.4	119.8	588.5		453.5	442.0	56.0	400.4		208.0		102.1	71.7	422.7
6-Methylhept-5-en-2-one	51.6	74.9	84.4	492.9	43.5	225.1	281.0	11.8	71.1	29.3	201.0		25.2	23.1	13.9
Butanol	1345.0	407.3	677.4	882.1	435.5	578.3	1502.4	436.0	1195.4	252.9	826.3	220.7	617.4	946.6	1704.0
Butyl 2-methylbutyrate	46.4	183.0	1143.0	912.3	99.3	474.7	843.2	117.4	1693.4	54.6	148.1	145.2	249.8		
Butyl acetate	253.7	685.4	9168.5	2340.6	1318.2	1062.1		5164.3	540.1	266.4	130.1	2046.8	10,945.3	7716.6	
Butyl butyrate	224.2	730.6	359.1	290.4	320.8	882.3	2542.0	2081.5	292.0	78.3	3073.6	163.0		787.9	3050.3
Butyl hexanoate	193.8	758.0	853.0			1532.2	292.1	1131.7	70.4		445.4	378.9	1025.0	1046.6	174.9
Butyl propanoate	179.5	776.6	135.2	250.4	148.3	516.5		255.0	1168.6	59.1	117.8	866.6	578.6		
Ethyl butyrate	143.1	2233.5		9102.3	751.7	307.8	4174.7			401.6	2518.2			3365.2	591.9
Ethyl hexanoate	200.5	449.1		1827.4	345.3	921.3	378.8	318.3	616.9	177.7	674.0			1099.7	
Ethyl octanoate	42.6			1309.1	76.7	889.4			448.3	84.7					
Hept-2-enal	11.9		17.8	15.7	10.3	77.9	23.4	4.6	6.9	27.5	16.8		9.4	6.7	
Heptanol		76.7	100.4	261.4		128.5	309.7	39.0	316.6	52.0	66.5	58.7	60.1	126.9	177.3
Hex-2-en-1-ol	216.3	202.0	200.1		252.6		286.1	63.8	93.6	419.1	164.1	330.4	276.0	322.1	851.6
Hex-2-enal	180.4	958.0	1854.2	439.9	1270.7	1070.4	733.3	948.4	695.3	1639.6	669.3	2724.6	3010.9	1294.0	2332.1
Hexanal	80.2	396.4	1198.0	369.3	380.4	226.8	474.9	98.1	232.5	177.7	146.9	353.8	404.9	327.8	299.3
Hexanol	1310.6	1568.6	2264.7	4686.9	1946.5	1968.3	2439.5	1196.3	1477.8	2015.8	2888.6	2149.4	2986.7	3014.3	1027.7
Hexyl 2-methylbutyrate	353.7	1671.4	786.0	316.0	258.6	352.5	258.2	816.1	2407.7	217.4	1168.6	562.2	1183.3	154.0	844.0
Hexyl acetate	30.0	435.8	16,434.0	3323.6	75.4	263.6		9791.7	573.5	31.3	118.1	4940.9	13,763.3	3607.2	
Hexyl butyrate			475.8	2580.4	364.8	2115.8	1166.1	1219.2	1480.3	335.4	865.3	184.1	1470.6	917.3	893.4
Hexyl hexanoate	80.7	215.9	430.5	579.4	51.9	383.9	604.8	660.0	613.8	63.2	198.3	1161.7	1496.9	119.9	959.5
Hexyl propanoate		188.6	169.5	529.1		146.4			941.1		973.4	644.7	262.3		
Methyl butyrate	290.1	272.2			111.4	1031.2					1013.8				
Methyl hexanoate	54.9	241.4		763.3	35.3	405.4		705.4			1340.7	216.6			
Pentanol	623.8	305.3	87.1	541.5	251.0	277.7	323.6	82.0	549.6	278.8	656.3	66.9	81.8	240.9	218.3
Propyl butyrate	401.8	298.8		142.5	1145.9	156.0	206.3		555.8	324.8	358.7	5940.9		262.4	368.6

## Data Availability

Data are contained within the article and Appendix A.

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
