# Peer review of "Fingerprinting of Volatile Organic Compounds in Old and Commercial Apple Cultivars by HS-SPME GC/GC-ToF-MS"

_ijms, 2024, doi:10.3390/ijms252413478_

Round 1
Reviewer 1 Report
Comments and Suggestions for Authors
The current submission identified the volatile organic compounds in 30 old and 7 commercial apple cultivars by HS-SPME GC/GC-ToF-MS, which is interesting to the readership of International Journal of Molecular Sciences. The experimental design and writing of the manuscript are good. I would suggest the authors improve the manuscript by considering the following suggestions.
1. The materials collected for the current study were from different locations for the old cultivars and commercial cultivars, which would affect the compounds. The authors are suggested to address the shortages and limitations of the study in Discussion section.
2. The current results of selected compounds clustering can distinguish the old and commercial cultivars, which is good for further commercial use. However, I would suggest the authors compare the previous evaluation methods with their current work to show the advantages and significance of the current study.
3. The citation format needs to be further checked and corrected. For example, Line 124: Wu et al., 2022 should be replaced by Wu et al. (2022), etc.
Author Response
Dear Reviewer
We are very grateful for the thorough reviews and the valuable suggestions. We have taken into account all of them. We believe that the introduced changes will be beneficial to the quality of the manuscript.
Q1: The materials collected for the current study were from different locations for the old cultivars and commercial cultivars, which would affect the compounds. The authors are suggested to address the shortages and limitations of the study in Discussion section.
A: Thank you for your suggestion. In total, the material came from 10 orchards, some of the old varieties also from these commercial orchards. Nevertheless, the old varieties stood out significantly. We have added the appropriate notes in the discussion.
Q2: The current results of selected compounds clustering can distinguish the old and commercial cultivars, which is good for further commercial use. However, I would suggest the authors compare the previous evaluation methods with their current work to show the advantages and significance of the current study.
A: Thanks for your suggestion. We have discussed the shortcomings of other studies that we have pointed out in the discussion. We have added an additional paragraph regarding the significance of our work to the previous state of knowledge on apple (and other agricultural products) fingerprinting.
Q3: The citation format needs to be further checked and corrected. For example, Line 124: Wu et al., 2022 should be replaced by Wu et al. (2022), etc.
A: Thank you. Corrected.
Reviewer 2 Report
Comments and Suggestions for Authors
The aim of this study was to determine whether apple cultivars can be fingerprinted by grouping varieties based on their volatile organic compound profiles. To achieve this, Headspace SPME and GC-GC-ToF-MS techniques were employed. A total of 30 traditional apple cultivars from the same germplasm, along with 7 commercial cultivars, were analyzed, resulting in the identification of 119 volatile organic compounds. However, several concerns need to be addressed before further consideration. Some comments are provided below.
· The abstract requires overall improvement; for example, it lacks a clearly stated objective and does not provide details about the methods used. Additionally, the structure and clarity could be enhanced to better convey the study’s key findings and significance. The objective stated in the introduction section also needs improvement.
· The results should be compared more extensively with the latest research findings.
· The researchers analyzed 30 different old cultivars and 7 commercially grown cultivars, with three repetitions, yet they also mentioned 87 individual apple fruit samples. Could you clarify how the total of 87 was calculated?
· Have you evaluated the detection limit of your analysis?
· Lines 305–308: The references cited here do not appear to be appropriate for the conclusion section. It may be unnecessary to include these studies at this point.
· Line 294: You mentioned nine investigations, but Table 1 includes more than nine studies. Could you clarify this discrepancy?
Author Response
Dear Reviewer
We are very grateful for the thorough reviews and the valuable suggestions. We have taken into account all of them. We believe that the introduced changes will be beneficial to the quality of the manuscript.
Q1: The abstract requires overall improvement; for example, it lacks a clearly stated objective and does not provide details about the methods used. Additionally, the structure and clarity could be enhanced to better convey the study’s key findings and significance. The objective stated in the introduction section also needs improvement.
A: Thank you for your suggestion. We have added more information in the abstract and introduction.
Q2: The results should be compared more extensively with the latest research findings.
A: We were unable to find studies on the quantitative analysis of VOCs from apples more recent than those pointed in the discussion (Wu et al., (2022) and Yang et al., (2021)).
Q3: The researchers analyzed 30 different old cultivars and 7 commercially grown cultivars, with three repetitions, yet they also mentioned 87 individual apple fruit samples. Could you clarify how the total of 87 was calculated?
A: There were 30 old varieties and 7 new ones, but among both the old and new ones there were some that came from several places of cultivation. Please look at the supplementary file - for example, the Gala variety was tested from seven places, and the Golden Delicious from three, etc.
Q4: Have you evaluated the detection limit of your analysis?
A: We did not calculate detection limits, because our intention was not to find as many compounds as possible, many of them close to the detection limit, but to analyze the more important compounds, allowing for fingerprinting. Even in the data processing method itself, we set the threshold relatively high, in order to get rid of these insignificant peaks from the results (which we emphasize in subchapter 3.6.).
Q5: Lines 305–308: The references cited here do not appear to be appropriate for the conclusion section. It may be unnecessary to include these studies at this point.
A: Thank you for your suggestion. We've moved this part to another section.
Q6: Line 294: You mentioned nine investigations, but Table 1 includes more than nine studies. Could you clarify this discrepancy?
A: Thank you for pointing out the discrepancy. Corrected.
Round 2
Reviewer 2 Report
Comments and Suggestions for Authors
Thank you to the authors for addressing my comments satisfactorily, except for one. I still believe the results should be compared more extensively with the latest research findings. The authors stated, "We were unable to find studies on the quantitative analysis of VOCs from apples more recent than those cited in the discussion (Wu et al., 2022, and Yang et al., 2021)." However, if recent studies are not available, they could also consider slightly older research. For instance, I easily found a relevant study published in 2023 related to the characterization of key volatile compounds in Qinguan apples using GC-MS. While they are not required to cite this specific reference, a more thorough search would likely uncover additional relevant studies.
Author Response
Dear reviewer,
Thank you for pointing out the appropriate article. We've added it to the manuscript and included it in the discussion. We found many articles that profile the composition of VOCs from apples, but almost all of them only provide percentages or peak areas. The one by Ruy Li et al (2023) we found only after your suggestion. Thank you again for your contribution to improving our manuscript.
Round 3
Reviewer 2 Report
Comments and Suggestions for Authors
The authors have made the necessary changes as requested.